# Exposure of Colon-Derived Epithelial Monolayers to Fecal Luminal Factors from Patients with Colon Cancer and Ulcerative Colitis Results in Distinct Gene Expression Patterns

**DOI:** 10.3390/ijms25189886

**Published:** 2024-09-13

**Authors:** Maria K. Magnusson, Anna Bas Forsberg, Alexandra Verveda, Maria Sapnara, Julie Lorent, Otto Savolainen, Yvonne Wettergren, Hans Strid, Magnus Simrén, Lena Öhman

**Affiliations:** 1Department of Microbiology and Immunology, Institute of Biomedicine, Sahlgrenska Academy, University of Gothenburg, 405 30 Gothenburg, Sweden; 2National Bioinformatics Infrastructure Stockholm (NBIS), Science for Life Laboratory, Department of Biochemistry and Biophysics, Stockholm University, 106 91 Stockholm, Sweden; 3Chalmers Mass Spectrometry Infrastructure, Department of Biology and Biological Engineering, Chalmers University of Technology, 412 96 Gothenburg, Sweden; 4Institute of Public Health and Clinical Nutrition, University of Eastern Finland, 70210 Kuopio, Finland; 5Department of Surgery, Institute of Clinical Sciences, Sahlgrenska Academy, University of Gothenburg, 413 45 Gothenburg, Sweden; 6Department of Surgery, Sahlgrenska University Hospital, Region Västra Götaland, 416 85 Gothenburg, Sweden; 7Department of Internal Medicine, Södra Älvsborg Hospital, 501 82 Borås, Sweden; 8Department of Molecular and Clinical Medicine, Institute of Medicine, Sahlgrenska Academy, University of Gothenburg, 413 45 Gothenburg, Sweden; 9Center for Functional GI and Motility Disorders, University of North Carolina, Chapel Hill, NC 27599, USA

**Keywords:** epithelial monolayers, epithelial barrier, intestinal microenvironment, inflammatory bowel disease, colon cancer

## Abstract

Microbiota and luminal components may affect epithelial integrity and thus participate in the pathophysiology of colon cancer (CC) and inflammatory bowel disease (IBD). Therefore, we aimed to determine the effects of fecal luminal factors derived from patients with CC and ulcerative colitis (UC) on the colonic epithelium using a standardized colon-derived two-dimensional epithelial monolayer. The complex primary human stem cell-derived intestinal epithelium model, termed RepliGut^®^ Planar, was expanded and passaged in a two-dimensional culture which underwent stimulation for 48 h with fecal supernatants (FS) from CC patients (*n* = 6), UC patients with active disease (*n* = 6), and healthy subjects (HS) (*n* = 6). mRNA sequencing of monolayers was performed and cytokine secretion in the basolateral cell culture compartment was measured. The addition of fecal supernatants did not impair the integrity of the colon-derived epithelial monolayer. However, monolayers stimulated with fecal supernatants from CC patients and UC patients presented distinct gene expression patterns. Comparing UC vs. CC, 29 genes were downregulated and 33 genes were upregulated, for CC vs. HS, 17 genes were downregulated and five genes were upregulated, and for UC vs. HS, three genes were downregulated and one gene was upregulated. The addition of FS increased secretion of IL8 with no difference between the study groups. Fecal luminal factors from CC patients and UC patients induce distinct colonic epithelial gene expression patterns, potentially reflecting the disease pathophysiology. The culture of colonic epithelial monolayers with fecal supernatants derived from patients may facilitate the exploration of IBD- and CC-related intestinal microenvironmental and barrier interactions.

## 1. Introduction

The interaction between the host and the local intestinal microenvironment, i.e., the microbiota and the metabolites the microbiota produces, seemingly play a central role in the development of non-communicable diseases in the gut [1]. A compromised epithelial barrier may facilitate the translocation of luminal microbiota and their metabolites, leading to loss of gut homeostasis, inflammation, and sustaining the pathogenesis of colon cancer (CC) and inflammatory bowel disease (IBD), including ulcerative colitis (UC) [2,3,4,5]. Colon cancer is a prevalent form of cancer and a major contributor to cancer-related deaths globally. This disease encompasses a varied group of tumors, each exhibiting distinct clinical and pathological characteristics and outcomes [6]. Ulcerative colitis (UC) is a long-term inflammatory condition of the colon and rectum. The disease has become a significant global health issue due to its high prevalence in developed nations and the rising incidence in developing countries [7]. While disease mechanisms for CC and UC are yet to be revealed, various factors, including genetic predisposition, environmental influences, luminal factors, and mucosal immune dysregulation, are believed to play a role in their development [5,8].

Over the past decade, research exploring the connection between intestinal diseases and deviations in gut microbiota has surged, especially highlighting compromised microbial diversity [9]. The findings underscore the intricate nature of these conditions, involving changes in the microbial community, dysfunction of the intestinal epithelial barrier, and altered immune responses [10,11]. However, current investigative strategies often focus separately on the characteristics of the immune response and microbiota composition, leaving a gap in the understanding of the intricate relationship between the local microenvironment, barrier integrity, and host immunity [12]. To address this gap, there is a growing demand for in vitro cell culture setups that assess the impact of the luminal content, particularly in the context of diseases. Such disease modeling aims to establish how the luminal environment influences intestinal epithelial homeostasis and explores potential cellular and molecular disease mechanisms, but can also serve as a relevant system for compound screening [13,14]. 

Our group has recently demonstrated that the stimulation of Caco-2 cells and colonic organoids from a healthy donor, with fecal supernatants derived from patients with different gastrointestinal diseases, resulted in distinct gene expression profiles, potentially reflecting the luminal microenvironment of the fecal sample donor [15,16]. This experimental approach allows for investigating the host-microbiota crosstalk at the epithelial barrier and the effects of the gut microenvironment in the pathogenesis of intestinal diseases. However, the work with organoids is technically challenging and time-consuming, and organoids are subject to genetic and environmental variability [17]. The Caco-2 cell line, comprising immortalized epithelial cells but lacking other cell types crucial for maintaining the functional properties of the intestinal epithelial barrier, has other limitations, such as intrinsic heterogeneity of the parental cell line and culture-related conditions influencing morphological and functional characteristics of the mature enterocyte [18]. Thus, there is a need for an experimental cell culture model that bypasses the need to establish and maintain organoids from intestinal biopsy donors, while utilizing a cell line more complex than Caco-2 or similar to it. By delving deeper into the interactions between the host and microbiota, experimental cell culture models could enhance our understanding of the development of intestinal diseases, potentially paving the way for new therapeutic options. Therefore, in this study, we exposed the standardized complex primary human stem cell-derived intestinal epithelium model termed RepliGut^®^ Planar, established from a healthy donor, to fecal supernatants obtained from patients with CC and UC as well as from healthy subjects. The objective was to establish an in vitro platform to discern the effects of luminal content on function and properties of the intestinal epithelium as a model system for gastrointestinal diseases. 

## 2. Results

### 2.1. Study Subjects, Study Samples, and Assessment of Transepithelial Electrical Resistance (TEER) in the Colon-Derived Two-Dimensional Epithelial Monolayers 

In this study, patients with UC (*n* = 6), CC (*n* = 6), and HS (*n* = 6) were included (for characteristics see Table 1). The healthy subjects were younger than the CC patients (*p* < 0.001). FS was prepared and LC/MS analysis displayed differences in the FS metabolite profiles between the groups (Figure 1A). The colon-derived two-dimensional epithelial monolayers were differentiated for 2 days after which the confluent cell layers were polarized as shown by surface staining of phospho-ezrin and actin filaments (Figure 1B). FS was added to the apical side of the transwells, followed by incubation for 48 h. The integrity of the monolayers was evaluated by TEER measurements at times 0 h, 24 h, and 48 h and only addition of TNFα to the basolateral compartment (inflammation control) altered the monolayer integrity (Figure 1C). 

### 2.2. Gene Expression in Epithelial Monolayer Is Altered by Fecal Supernatants (FS) and Linked to Each Patient Group 

To investigate the effect of FS on gene expression, mRNA sequencing was performed and a principal component analysis (PCA) of all genes for all samples revealed the most pronounced effect for the TNFα treated monolayers, but also alterations for FS treated as compared to untreated monolayers (Figure 2A). Next, the controls, i.e., the TNFα treated and untreated monolayers, were excluded and the new PCA showed tendencies to group-clustering for HS, UC, and CC (Figure 2B). Clustering patterns were also explored by a heatmap of distances between samples, and the samples of the healthy group showed high gene expression similarities, while UC and CC were grouped together (Figure 2C). 

Next, low expression genes were filtered out by excluding genes with two or less samples having at least 10 counts. The new dataset was evaluated for CC vs. HS, UC vs. HS, and UC vs. CC. For CC vs. HS, 17 genes were significantly downregulated, and 5 genes were significantly upregulated at an FDR threshold of 0.05 (Figure 3A, Table 2). The differential expression results for the four genes with lowest *q*-values were DDB1 and CUL4 associated factor 8 (*DCAF8*), pellino E3 ubiquitin protein ligase family member 2 (*PELI2*), cyclin Y (*CCNY*), and low-density lipoprotein receptor-related protein 6 (*LRP6*) (Figure 3B). 

For UC vs. HS, three genes were significantly downregulated, and one gene was significantly upregulated at an FDR threshold of 0.05 (Figure 4A, Table 3). The four differentially expressed genes were PBX homeobox interacting protein 1 (*PBXIP1*), 17β-Hydroxysteroid dehydrogenase2 (*HSD17B2*), tubulin alpha 1b (*TUBA1B*), and cerebral endothelial cell adhesion molecule (*CERCAM*) (Figure 4B). 

Finally, for UC vs. CC, 29 genes were significantly downregulated, and 33 genes were significantly upregulated at an FDR threshold of 0.05 (Figure 5A, Table 4). The differential expression results for the four genes with lowest *q*-values were *DCAF8*, gigaxonin (*GAN*), *PELI2*, and coagulation factor VIII associated 1 (*F8A1*) (Figure 5B). 

### 2.3. Epithelial Monolayer Secretion of IL8 Is Altered by Fecal Supernatants 

Finally, we investigated how FS from HS, CC patients or UC patients influenced cytokine secretion from the monolayers. While no differences between the groups were detected for IL1β, IL8, or TNFα expression (Figure 6A), a PCA of all cytokines displayed tendencies for group-clustering for the three study groups compared to the untreated monolayers (Figure 6B). When comparing cytokine expression from all FS treated monolayers (the three study groups, *n* = 18) vs. untreated cells (*n* = 3), FS was shown to induce the secretion of IL8 (719 (291–1180) vs. 580 (488–600) pg/mL (median (range)), *p* = 0.03). 

## 3. Discussion

In this investigation, the differences in metabolite compositions in fecal supernatants between patients with CC and UC led to distinct gene expression profiles when cultivated with colon-derived epithelial monolayers. Consequently, utilizing in vitro cell culture models exposed to fecal supernatants emerges as a promising method to simulate interactions between the intestinal epithelium and luminal content, providing valuable insights into gut barrier crosstalk in the context of intestinal diseases.

When adding FS from CC and UC patients to monolayers, the most pronounced effect was reduced expression of multiple genes, and more genes were downregulated, as compared to monolayers exposed to FS from HS. The reduced expression of specific genes of the monolayers suggests loss of function or impaired regulatory properties, which are traits commonly attributed to the epithelial barrier [19] in gastrointestinal diseases. For example, profound changes in gene expression in the colonic epithelium of patients with active UC have been reported [20], with more genes being downregulated than upregulated. Further, 17 genes were found to be consistently downregulated over five different CC patient datasets [21]. Thus, the downregulation of several genes in our in vitro cell culture model reflects changes in intestinal gene expression in CC as well as UC.

In more detail, FS from CC patients, but not UC patients, reduced the monolayer expression of *DCAF8*, *PELI2*, *CCNY,* and *LRP6*. *DCAF8* is an epigenetic modulator of ferroptosis but also regulates the function of myeloid leukemia factor [22], which has been linked to various forms of cancers. *PELI2* controls activation of the NLRP3 inflammasome and *PELI2* deficient mice have impaired response to toll-like receptor priming, NLRP3 stimuli, and bacterial challenge [23]. *CCNY* and *LRP6* are key players in the activation of the WNT/β-catenin signaling pathway [24] and, consequently, in the regulation of tissue homeostasis under physiological and pathological conditions. In contrast, the addition of FS from UC patients, but not CC patients, reduced the monolayer expression of *PBXIP1, HSD17B2,* and *TUBA1B*. *PBXIP1* plays a vital role in stem cell development and is elevated in rapidly proliferating cells, regulating cell cycle checkpoints [25]. Decreased expression of *HSD17B2*, an enzyme catalyzing steroid hormones and maintaining hormone balance, appears to be a frequent feature in non-small cell lung cancer [26,27]. *TUBA1B* enables double-stranded RNA binding activity and ubiquitin protein ligase binding activity, and low *TUBA1B* expression has been associated with adverse effects on the overall survival of patients with colon adenocarcinoma [28]. Thus, the addition of FS from CC and UC patients influences the expression of several different genes shown to be of importance for cell-signaling, inflammation, and tumorigenesis. However, none of the genes found to be differentially expressed in monolayers after the addition of FS from CC and UC patients, respectively, overlapped, and we conclude that FS from the two patient groups have different effects on the monolayer gene expression. Although on a somewhat different note, differently expressed fecal as well as serum miRNAs have been identified for CC as well as UC patients, providing a potential source for biomarkers but also reflecting the subtle regulation of gene transcription associated to diseases of the gastrointestinal tract [29,30].

Gut metabolites serve as a dynamic reflection of the luminal environment, host behaviors, and a manifestation of microbial activity, transcending diversity fluctuations observed in microbiota-centric studies [12]. Previous reports highlighting differences in the metabolite composition of fecal samples from UC and CC patients compared to HS align with our findings, reinforcing the notion that luminal content is disease-specific [31,32]. In our study, an untargeted metabolite analysis unveiled distinct patterns in fecal metabolite compositions between patient groups and healthy subjects. While our chosen metabolite analysis method did not include compound annotation for potential disease-specific biomarker identification, the results suggest that fecal metabolite composition analysis could be developed into a non-invasive diagnostic tool for gastrointestinal diseases in the future. 

The addition of FS on the apical side of the differentiated polarized monolayers did not endanger the integrity of the epithelial layer. However, the addition of TNFα to the basolateral compartment, used as a positive control mimicking inflammation, disrupted the barrier integrity, shown by loss of TEER. Therefore, the TNFα stimulated monolayers were excluded from further analyses. Addition of FS to monolayers induced a substantial secretion of IL8, whereas the effect on TNFα and IL1β secretion was much less pronounced. The cytokine secretion mirrored the gene expression, and gene counts for IL8 were in general 10–100 times higher than gene counts for TNFα and IL1β, likely reflecting activated signaling pathways and the dynamics of cytokine expression in epithelial cells, but also the relatively short cell culture time (48 h). Previous research has established a connection between intestinal microbiota and IL8 production, suggesting that different microbial compositions can influence the production of cytokines like IL8 [33], which aligns with our findings. Interleukin-8 is a chemoattractant cytokine specifically targeting neutrophils and plays a crucial role in attracting and activating these cells in areas of inflammation [34]. Still, in contrast to gene expression, the effects of FS on cytokine secretion did not differ between study groups, and we speculate that a prolonged experimental time frame could support distinct cytokine profiles reflecting the disease state of the FS donor.

We acknowledge that human-derived metabolites, apart from microbiota-derived ones, could influence fecal metabolite profiles, although this aspect was not explored due to the complex and challenging task of separating these entities. On the same note, we cannot rule out the possibility that other factors present in FS, such as proteins, lipids, or various PAMPs, might be responsible for the effects observed in our in vitro culture model. However, we have previously shown that lipopolysaccharide levels did not differ between FS from different study groups [16]. The diversity and resulting influence of the metabolite composition of FS on monolayers may indeed be linked to the individual microbiota of the donor of fecal material, and as the intestinal microenvironment varies with age [9], the age differences among study groups could have influenced the observed gene expression differences. Study participants provided only a single fecal sample, which, although consistent with the observed stability of gut microbiota and fecal metabolome over time [35], may not capture dynamic variations. Further, a significant limitation to the study is the small sample size within each group, which diminishes statistical power. This limitation may potentially reduce the ability to detect subtle yet physiologically relevant differences between groups, leading to underestimation of the effects caused by the FS. By having lager sample sizes in future studies, the generated data would be more robust, leading to more reliable conclusions. Finally, the use of colon-derived epithelial monolayers from a single healthy donor introduces another potential drawback. The unique genetic characteristics of the donor may influence the results, limiting the generalizability of the findings. Relying on a single donor introduces a limitation in accounting for inter-individual variability, which may lead to overlooking significant differences in how different individuals respond to similar stimuli. 

In summary, treatment of colon-derived epithelial monolayers with FS originating from CC and UC patients induces changes in gene expression profiles, potentially reflecting the luminal microenvironment of the respective fecal sample donors. The experimental methodology outlined in this study offers promise for advancing our comprehension of how environmental factors contribute to the pathogenesis of intestinal diseases. By further exploring the influences of the host-microbiota crosstalk, this experimental approach could contribute to a deeper understanding of intestinal disease pathogenesis, potentially leading to novel therapeutic opportunities.

## 4. Materials and Methods

### 4.1. Study Subjects and Sample Collection

Patients with histologically diagnosed colon cancer awaiting surgery, patients with histologically verified diagnosis of UC, and healthy subjects were recruited at the Sahlgrenska University hospital, Gothenburg, Sweden. Fecal samples from the CC patients were collected immediately prior to tumor resection surgery, the tumor was subsequently staged according to TNM international classification [36]. Fecal samples from the patients with UC were collected during a disease flareup, defined as an endoscopic Mayo score of ≥2 and a total Mayo score of ≥3, according to international criteria [37]. Healthy subjects had no current or prior history of gastrointestinal or other chronic disorders, nor had they taken any medication, including immunosuppressive agents, during the 3 months before sample collection. Study subjects collected fecal samples at home and kept them in the freezer until transportation. Samples were then stored on site at −80 °C until preparation of fecal supernatants. All subjects gave written informed consent prior to participation in the corresponding studies. The study was conducted in accordance with the Declaration of Helsinki and approved by the Regional Ethical Review Board at the University of Gothenburg (Dnr 563-02, date of approval 19 November 2002; Dnr 233-10, date of approval 17 August 2010; Dnr 988-14, date of approval 26 February 2015).

### 4.2. Preparation of Fecal Supernatants

Feces were weighed and dissolved in two weight volumes of ice-cold phosphate-buffered saline (PBS), prior to centrifugation for 10 min at 1600× *g*. The liquid phase was then ultra-centrifuged at 35,000× *g* for 2 h at 4 °C. The collected fecal supernatant was stored at −80 °C until use. 

### 4.3. Liquid Chromatography−Mass Spectrometry

Metabolites of the fecal supernatants were analyzed at Chalmers Mass Spectrometry Infrastructure (Gothenburg, Sweden), using a non-targeted liquid chromatography−mass spectrometry (LC/MS) approach, as described in detail by Holst et al. [15]. The analyses involved reversed-phase chromatography and hydrophilic interaction chromatography with positive and negative electrospray ionization [38]. Samples from each study group were analyzed in separate batches, each with its own quality control samples. The “notame” analytical workflow, as outlined by Zheng et al. [38], was employed for data pre-processing, including drift correction within and between batches. Data imputation was conducted using the missForest R package [39], and feature clustering was performed to eliminate weak and repeated features [38]. Log10 transformation was applied before between-batch correction to minimize potential batch effects caused by the instrument.

### 4.4. Stimulation of Colonoid Monolayers with Fecal Supernatants

Primary human intestinal cells expanded as monolayers, RepliGut^®^ Planar models, (RepliGut system, Altis Biosystems, Durham, NC, USA) were allowed to differentiate and grow to confluency according to the manufacturer’s instructions before being cultured for 48 h at 37 °C, 5% CO_2_, with FS from HS, UC patients, or CC patients. The FS were diluted 1:100 [12] and added to the apical surface of the monolayers. For negative controls (*n* = 2), an equal volume of differentiation media was added to generate non-stimulated monolayers. Tumor necrosis factor-alpha (TNFα) (Sigma-Aldrich, Saint Louis, MO, USA) diluted in differentiation media to a final concentration of 50 ng/mL in the basolateral compartment, was used as pro-inflammatory, positive control (*n* = 2). Additionally, monolayers were incubated with media alone for microscopy. 

### 4.5. Immunofluorescence and Imaging

Immunofluorescence staining of RepliGut^®^ Planar monolayers was performed against phospho-ezrin in combination with phalloidin and Hoechst. Monolayers were stained with rabbit anti-phospho-Ezrin IgG (dilution 1:200; Abcam, Cambridge, UK) and subsequently with anti-rabbit Alexa 488 (dilution 1:200; Life Technologies, Carlsbad, CA, USA) to visualize polarized intestinal epithelial cells. Thereafter, incubation was followed with Phalloidin-647 (dilution 1:500, Abcam) to visualize actin filaments, and finally with Hoechst 33342 (dilution 1:10,000, ThermoFisher Scientific, Waltham, MA, USA) to visualize cell nuclei. Washes with PBS buffer followed every step described. Stained membranes were mounted with ProLong Diamond Antifade Mountant (ThermoFisher Scientific) on a glass slide. The preparations were kept sealed at 4 °C until visualization. Images of stained monolayers were acquired using 20× and/or 40× objective and the same acquisition settings on an LSM 700 inverted confocal microscope (Carl Zeiss, Oberkochen, Germany) using Zen 2012 SP5 (Black edition, version 14.0.3.201) imaging software. Confocal images were processed using the software Zen 3.0 2019 (Blue version) and Fiji (ImageJ version 1.52p). Changes in brightness/contrast and reduction in background noise were applied to emphasize the qualitative analysis of orthogonal images, respectively. 

### 4.6. Gene Expression Analysis

Total RNA was isolated with the AllPrepR RNA/Protein kit (Qiagen, Hilden, Germany) following the manufacturer’s instructions. RNA was stored at −80 °C until analysis. Sequencing libraries from mRNA were prepared using Illumina TruSeq Stranded mRNA kit (National Genomic Infrastructure (NGI), SciLifeLab, Stockholm, Sweden) and RNA was thereafter sequenced using TaKaRa SMARTer pico RNA kit (NGI, Stockholm, Sweden).

### 4.7. Cytokine Analysis

Medium from the basolateral compartment of the monolayers was sampled and stored at −80 °C until analysis. Cytokine concentrations of IL1β, IL8, and TNFα were measured using the MSD^®^ Multi-Spot Assay System V-PLEX™ Proinflammatory Panel II Plus kit (Meso Scale Discovery, Rockville, MD, USA) following the manufacturer’s instructions.

### 4.8. Data Analysis

Bacterial contamination was assessed using FastQ Screen [40] version 0.14.0. Preprocessing of sequencing reads was performed using the nf-core/rnaseq pipeline [41] version 3.9 mapping to genome GRCh37. The library preparation protocol (SMARTer Total Stranded RNA-seq, Pico input mammalian—V3) included UMIs which were extracted and used to deduplicate reads using pipeline parameters –with_umi –umitools_extract_method “regex” –umitools_bc_pattern2 “^(?P.{8})(?P.{6}).*” rRNA reads were excluded using –remove_ribo_rna. Gene expression data quality was assessed using centered principal component analyses and heatmap of Euclidean distances between samples performed on variance stabilizing transformed data [42]. All available (sequenced) samples were kept in the analysis. Genes with very low expression were excluded and the differential expression was performed on the raw count data from the samples. DESeq2 version 1.38.1 was used for differential expression analyses [43]. The considered contrasts were: CC vs. HS; UC vs. HS, and UC vs. CC. Given that a filtering based on low expression was already performed, no additional independent filtering was performed in DESeq2. For genes to be statistically significant, a threshold at 0.05 on adjusted *p*-values was considered. The *p*-values were adjusted using the Benjamini–Hochberg method [44]. For top significant genes of each contrast, normalized counts were plotted for visualizing results. All analyses were performed in R version 4.3.0 (21 April 2023) [45].

Mann Whitney U test was used to evaluate differences between two groups and Kruskal–Wallis test followed by Dunn’s multiple comparisons test was used to evaluate differences between three groups. Analyses were performed using GraphPad Prism 10.0.2 (GraphPad Software, La Jolla, CA, USA); *p*-values < 0.05 were considered as statistically significant.

## Figures and Tables

**Figure 1 ijms-25-09886-f001:**
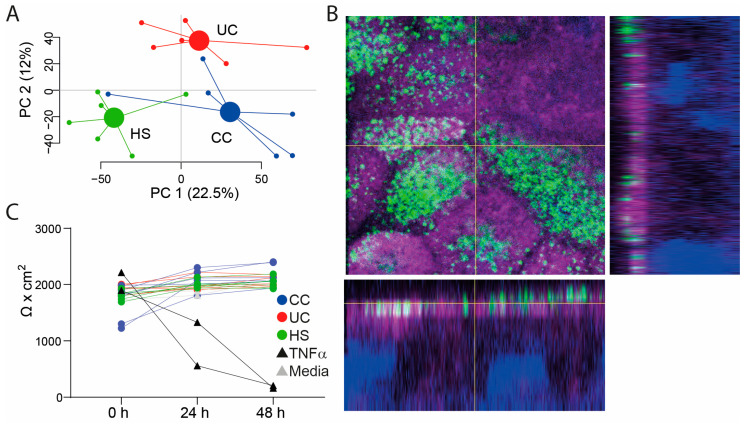
Fecal metabolite profiles and characterization of the RepliGut^®^ Planar monolayers treated with fecal supernatants (FS). (**A**) A principal component analysis based on 9699 spectral features detected in FS, analyzed by untargeted liquid chromatography/mass spectrometry, for healthy subjects (HS, *n* = 6, green dots), patients with ulcerative colitis (UC, *n* = 6, red dots) and colon cancer patients (CC, *n* = 6, blue dots). (**B**) Primary human intestinal cells in the RepliGut^®^ Planar system form a polarized monolayer with an apical membrane morphologically comparable with that of human intestine, as visualized by apical localization of phospho-ezrin (green), and actin filaments as detected by fluorescent phalloidin (magenta). Nuclei are visualized in blue color. The right hand and bottom panels show the orthographic view of the region, where “XY” and “XZ” indicate different cross-sections. The image was acquired with LSM700 inverted confocal microscope; 63× magnification. (**C**) Transepithelial electrical resistance was measured before and at 24 h and 48 h after addition of FS (HS: green dots, UC: red dots, CC: blue dots), TNFα (black triangles) or untreated (Media, gray triangles).

**Figure 2 ijms-25-09886-f002:**
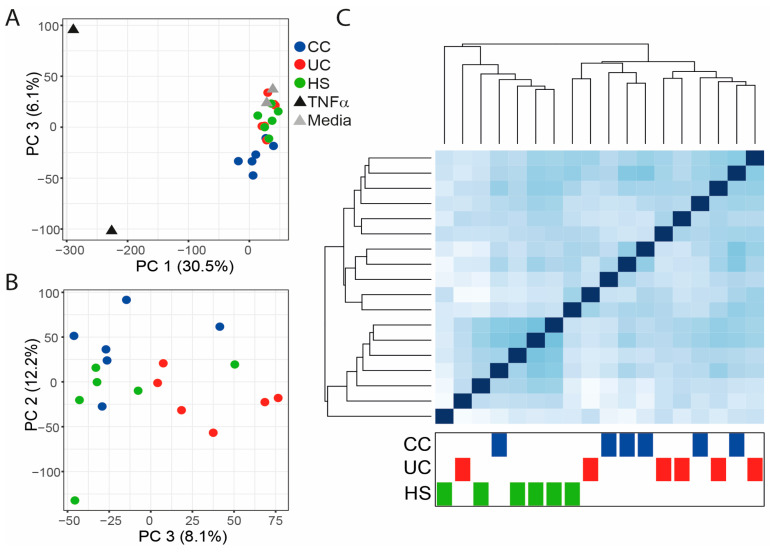
Gene expression of RepliGut^®^ Planar monolayers stimulated with fecal supernatants (FS) from healthy subjects (HS), patients with ulcerative colitis (UC), and patients with colon cancer (CC). Differentiated monolayers were stimulated apically with FS, TNFα, or left untreated (Media), for 48 h. Gene expression was analyzed by mRNA sequencing. (**A**) Principal component analysis (PCA) for monolayers treated with FS from HS, UC, CC, or with TNFα or media. (**B**) PCA for monolayers treated with FS from HS, UC, or CC. (**C**) Heatmap of distances between samples for monolayers treated with FS from HS, UC, or CC. HS *n* = 6 (green dots), UC *n* = 6 (red dots), CC *n* = 6 (blue dots), TNFα *n* = 2 (black triangles), and media *n* = 2 (gray triangles).

**Figure 3 ijms-25-09886-f003:**
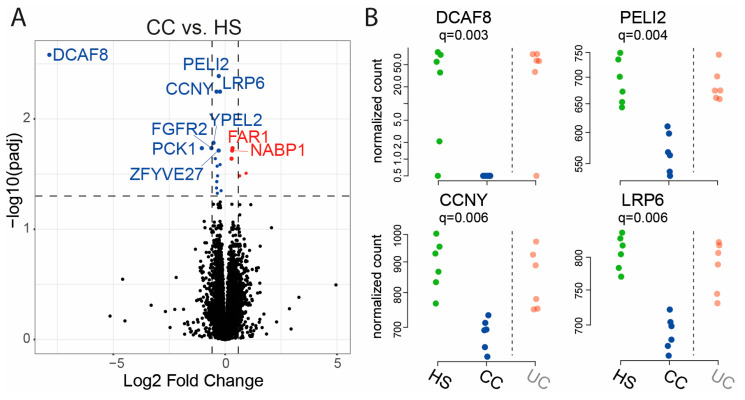
Comparison of gene expression of RepliGut^®^ Planar monolayers stimulated with fecal supernatants (FS) from patients with colon cancer (CC) and healthy subjects (HS). Differentiated monolayers were stimulated apically with FS for 48 h. Gene expression was analyzed by mRNA sequencing. (**A**) Volcano plot showing log2 fold change vs. significance. Wald test and false discovery rate analysis using the Benjamini−Hochberg method were used. Downregulated genes are shown in blue, upregulated genes in red and all the other genes in black. Horizontal dotted lines show cut-off for significance (q < 0.05) and vertical dotted lines show two-fold up- and downregulation. (**B**) Gene expression of the four most significant genes from (**A**). HS *n* = 6 (green dots) and CC *n* = 6 (blue dots). Results for patients with ulcerative colitis (UC) *n* = 6 (pale red dots) are shown to the right of the dotted line for reference.

**Figure 4 ijms-25-09886-f004:**
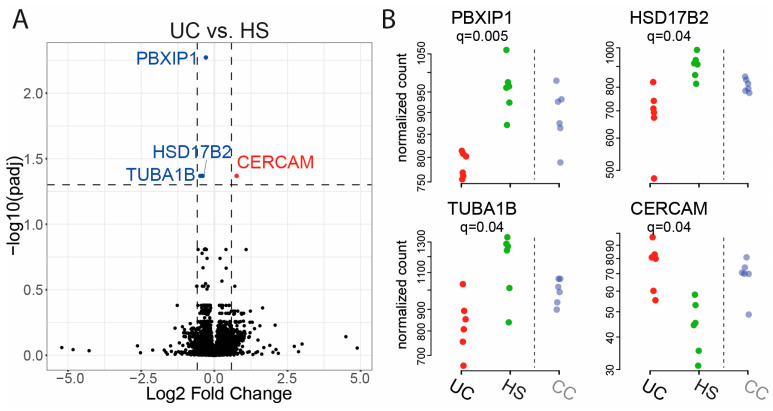
Comparison of gene expression of RepliGut^®^ Planar monolayers stimulated with fecal supernatants (FS) from patients with ulcerative colitis (UC) and healthy subjects (HS). Differentiated monolayers were stimulated apically with FS for 48 h. Gene expression was analyzed by mRNA sequencing. (**A**) Volcano plot showing log2 fold change vs. significance. Wald test and false discovery rate analysis using the Benjamini−Hochberg method were used. Downregulated genes are shown in blue, upregulated genes in red and all the other genes in black. Horizontal dotted lines show cut-off for significance (q < 0.05) and vertical dotted lines show two-fold up- and downregulation. (**B**) Gene expression of the four significant genes from (**A**). HS *n* = 6 (green dots) and UC *n* = 6 (red dots). Results for patients with colon cancer (CC) *n* = 6 (pale blue dots) are shown to the right of the dotted line for reference.

**Figure 5 ijms-25-09886-f005:**
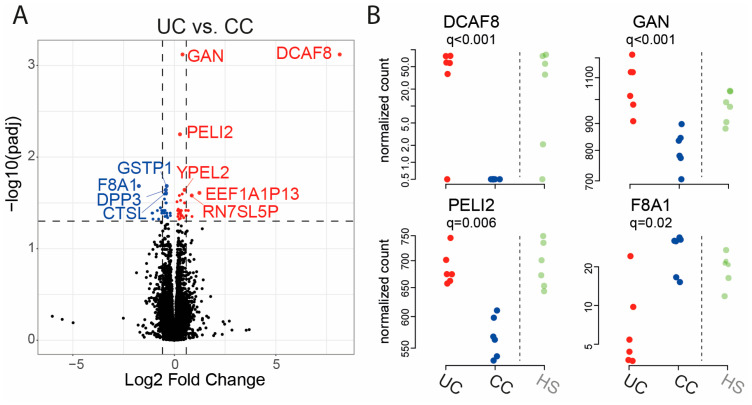
Comparison of gene expression of RepliGut^®^ Planar monolayers stimulated with fecal supernatants (FS) from patients with ulcerative colitis (UC) and patients with colon cancer (CC). Differentiated monolayers were stimulated apically with FS for 48 h. Gene expression was analyzed by mRNA sequencing. (**A**) Volcano plot showing log2 fold change vs. significance. Wald test and false discovery rate analysis using the Benjamini−Hochberg method were used. Downregulated genes are shown in blue, upregulated genes in red and all the other genes in black. Horizontal dotted lines show cut-off for significance (q < 0.05) and vertical dotted lines show two-fold up- and downregulation. (**B**) Gene expression of the four most significant genes from (**A**). UC *n* = 6 (red dots) and CC *n* = 6 (blue dots). Results for healthy subjects (HS) *n* = 6 (pale green dots) are shown to the right of the dotted line for reference.

**Figure 6 ijms-25-09886-f006:**
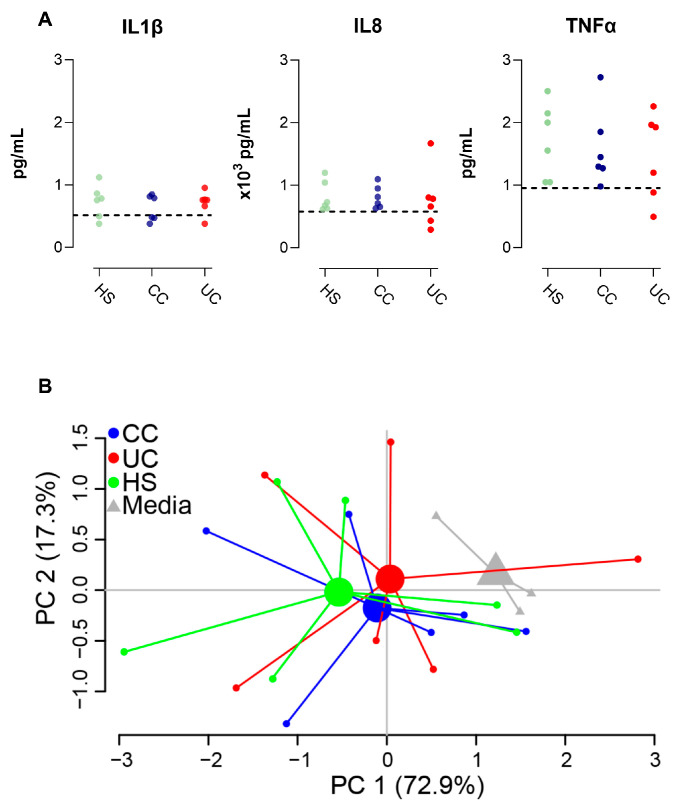
Cytokine secretion from RepliGut^®^ Planar monolayers stimulated with fecal supernatants (FS) from patients with ulcerative colitis (UC), patients with colon cancer (CC), and healthy subjects (HS). Differentiated monolayers were stimulated apically for 48 h with FS from HS, CC, UC, or media alone (*n* = 3). IL1β, IL8, and TNFα levels in the basolateral compartment were analyzed by MSD^®^ V-PLEX platform system. (**A**) Levels of IL1β, IL8, and TNFα. Dashed lines represent median cytokine concentration from monolayers cultured with media alone. (**B**) Principal component analysis based on the levels of IL1β, IL8, and TNFα. HS *n* = 6 (green dots), UC *n* = 6 (red dots), CC *n* = 6 (blue dots), and media *n* = 3 (gray triangles).

**Table 1 ijms-25-09886-t001:** Demographics and disease characteristics of study subjects.

	Ulcerative Colitis (*n* = 6)	Colon Cancer (*n* = 6)	Healthy (*n* = 6)
Age, median (range)	49 (40–67)	81 (68–91)	32 (25–44)
Sex, female/male	3/3	3/3	2/4
Mayo score, median (range)	9 (6–11)	N/A	N/A
Tumor stage ^1^, I/II/III/IV	N/A	2/1/2/1	N/A

Note: Abbreviations: N/A, not applicable. ^1^ According to TNM international classification.

**Table 2 ijms-25-09886-t002:** List of differentially expressed genes (*n* = 22) for RepliGut^®^ Planar monolayers stimulated with fecal supernatants (FS) from patients with colon cancer (CC) and healthy subjects (HS).

Gene Name	Fold Change	*q*-Value (CC vs. HS)
*CCNY*	−0.3857	0.006
*CNST*	−0.3386	0.033
*DCAF8*	−7.8589	0.003
*FAR1*	0.3316	0.018
*FGFR2*	−0.6135	0.018
*FLRT3*	0.9376	0.031
*GLDC*	0.6479	0.033
*GOPC*	0.2851	0.023
*KCNK5*	−0.3865	0.042
*LRP6*	−0.2310	0.006
*MBD1*	−0.3579	0.027
*MPZL3*	−0.3737	0.037
*NABP1*	0.3126	0.019
*PCK1*	−1.0437	0.018
*PELI2*	−0.2855	0.004
*PNCK*	−0.3762	0.037
*PPAP2B*	−0.4411	0.023
*PTK2B*	−0.1826	0.045
*RIN2*	−0.2263	0.026
*VEGFA*	−0.3487	0.047
*YPEL2*	−0.5200	0.016
*ZFYVE27*	−0.2907	0.019

Differentiated RepliGut^®^ Planar monolayers stimulated with FS from CC patients and HS. Diluted FS (1:100) was added apically to the monolayers and cultured for 48 h. Gene expression was analyzed by mRNA sequencing. Data show log2 fold change and *q*-values derived using Wald test adjusted by the Benjamini−Hochberg method.

**Table 3 ijms-25-09886-t003:** List of differentially expressed genes (*n* = 4) for RepliGut^®^ Planar monolayers stimulated with fecal supernatants (FS) from patients with ulcerative colitis (UC) and healthy subjects (HS).

Gene Name	Fold Change	*q*-Value (UC vs. HS)
*CERCAM*	0.7624	0.043
*HSD17B2*	−0.4005	0.043
*PBXIP1*	−0.2895	0.005
*TUBA1B*	−0.4825	0.043

Differentiated RepliGut^®^ Planar monolayers stimulated with FS from UC patients and HS. Diluted FS (1:100) was added apically to the monolayers and cultured for 48 h. Gene expression was analyzed by mRNA sequencing. Data show log2 fold change and *q*-values derived using Wald test adjusted by the Benjamini−Hochberg method.

**Table 4 ijms-25-09886-t004:** List of differentially expressed genes (*n* = 62) for RepliGut^®^ Planar monolayers stimulated with fecal supernatants (FS) from patients with colon cancer (CC) and patients with ulcerative colitis (UC).

Gene Name	Fold Change	*q*-Value (UC vs. CC)
*AL139819.1*	0.3827	0.044
*ALDH1A3*	−0.5806	0.038
*APLF*	−0.7745	0.048
*AVL9*	0.1957	0.038
*CCL2*	−1.0974	0.041
*CCNY*	0.3068	0.041
*CD63*	−0.1948	0.044
*CHD4*	−0.2044	0.041
*CLIP1*	0.1637	0.038
*CLN8*	0.4625	0.046
*CRHR1-IT1*	0.4987	0.032
*CTSL*	−0.4901	0.025
*DCAF8*	8.1459	0.001
*DNAJC11*	−0.4601	0.045
*DNAJC3*	0.1745	0.045
*DPP3*	−0.4107	0.025
*DYNC1LI1*	−0.5180	0.038
*EDIL3*	−1.0717	0.041
*EEF1A1P13*	1.2351	0.025
*ERN1*	0.2983	0.041
*F8A1*	−1.7489	0.021
*FUCA2*	−0.3988	0.032
*GAN*	0.4025	0.001
*GBA*	−0.3971	0.025
*GLDC*	−0.6223	0.038
*GREM1*	−0.8350	0.038
*GSTP1*	−0.3766	0.021
*HPGD*	−0.3718	0.044
*INHBA*	−1.0509	0.048
*INSIG1*	0.2298	0.038
*KANSL1*	0.2560	0.041
*KLF7*	0.2948	0.038
*KLHL28*	0.3201	0.044
*LRIG1*	0.5803	0.044
*LRP6*	0.1937	0.037
*LXN*	−0.4863	0.029
*MLLT4*	0.1303	0.031
*MLXIPL*	0.6879	0.038
*MSL1*	0.2348	0.048
*NCL*	−0.4204	0.023
*NEAT1*	0.5613	0.038
*PELI2*	0.2772	0.006
*PLA2G4C*	−0.5980	0.041
*PPP1R15B*	0.2462	0.026
*PRAP1*	−0.4953	0.041
*PTGES*	−0.6840	0.036
*RAPGEFL1*	0.3226	0.044
*RN7SL5P*	0.8572	0.026
*RP11-34P13.13*	0.8704	0.045
*RP11-395B7.2*	0.3212	0.038
*SEPP1*	−0.4565	0.038
*SH3D21*	0.3871	0.025
*SLC25A25*	0.1949	0.042
*SQSTM1*	−0.3686	0.041
*TM4SF20*	−0.4150	0.032
*TMEM176A*	−0.4591	0.038
*TMEM176B*	−0.3301	0.046
*TRIB3*	0.3095	0.030
*VEGFA*	0.3470	0.038
*YPEL2*	0.5035	0.023
*ZNF330*	−0.6198	0.041
*ZNF488*	0.2957	0.047

Differentiated RepliGut^®^ Planar monolayers stimulated with FS from CC and UC patients. Diluted FS (1:100) was added apically to the monolayers and cultured for 48 h. Gene expression was analyzed by mRNA sequencing. Data show log2 fold change and *q*-values derived using Wald test adjusted by the Benjamini−Hochberg method.

## Data Availability

The datasets used in this publication are available at https://doi.org/10.5878/bdc2-kk22 (published 26 August 2022).

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
