# Peer review of "Exposure of Colon-Derived Epithelial Monolayers to Fecal Luminal Factors from Patients with Colon Cancer and Ulcerative Colitis Results in Distinct Gene Expression Patterns"

_ijms, 2024, doi:10.3390/ijms25189886_

Round 1

Reviewer 1 Report

Comments and Suggestions for Authors

ijms-3195523

This study conducted by Magnusson et al. was aimed to determine the effects of fecal supernatants obtained from the patients of colon cancer (n=6) and ulcerative colitis (n=6), and healthy subjects (n=6) on the integrity, several genes’ expression, and several cytokines’ expression of the colon-derived epithelial monolayer. Although addition of fecal supernatants for 48 hours did not impair the integrity of the colonic epithelial monolayer, for colon cancer patients the treatment upregulated 5 genes and downregulated 17 genes, and for ulcerative colitis patients 1 gene upregulated and downregulated 3 genes, when comparing to healthy subjects. I think the results described are of interest. Several concerns must be reconsidered before publication.

1)     Numbers of samples are small. If possible, at least 10 samples from each disease (CC, UC, and HS) are needed for making conclusion.

2)     In this context, please shorty describe the limitations of this study in the Discussion section.

3)     Please specify “IL1” in the text and figure legend. IL1beta or IL1alpha?

4)     I guess the treatment may significantly alter the microenvironment and cytokines’ expression. However, in the study only IL8 was changed by the treatment. The reason for this can be described. Also, the role of IL8 can be mentioned and discussed.

5)     If possible, please cite one paper (PMID: 3864192).

Author Response

Dear Reviewer, please find the attached file containing our response.

Reviewer 2 Report

Comments and Suggestions for Authors

This is an original concerning the exposure of colon-derived epithelial monolayers to fecal luminal factors from patients with colon cancer and ulcerative colitis

The abstract is too generic and should be improved by adding the period of the study and underlining the results (with p-value included)

key results cannot be used in the abstract. just results

Please add the clinical implications of the study. Indeed, it is not clear how to apply the results of this study and their clinical significance

The limitations of the study must be underlined. For example: small sample size. Indeed, the article is similar to a short communication

The proper checklist (see equator network) must be added

The references on the topic should be updated and used in the discussion:

A Fecal MicroRNA Signature by Small RNA Sequencing Accurately Distinguishes Colorectal Cancers: Results From a Multicenter Study. Gastroenterology. 2023 Sep;165(3):582-599.e8. doi: 10.1053/j.gastro.2023.05.037.

A plasma microRNA panel for detection of colorectal adenomas: a step toward more precise screening for colorectal cancer. Ann Surg. 2013 Sep;258(3):400-8. doi: 10.1097/SLA.0b013e3182a15bcc

many others

A better introduction of both pathologies, especially UC, is necessary

Comments on the Quality of English Language

Minor

Author Response

(The authors gave the same response as above.)

Round 2

Reviewer 2 Report

Comments and Suggestions for Authors

I'm satisfied with the changes made

Comments on the Quality of English Language

minor editing before publication